# Safe reopening of college campuses during COVID-19: The University of California experience in Fall 2020

Brad H. Pollock[1]*, A. Marm Kilpatrick[2], David P. Eisenman[3,4], Kristie L. Elton[5], George W. Rutherford[6], Bernadette M. Boden-Albala[7], David M. Souleles[7], Laura E. Polito[8¤], Natasha K. Martin[9], Carrie L. Byington[5,10]

1 Department of Public Health Sciences, University of California Davis, Davis, California, United States of America, 2 Department of Ecology and Evolutionary Biology, University of California Santa Cruz, Santa Cruz, California, United States of America, 3 Division of General Internal Medicine and Health Services Research, David Geffen School of Medicine, Fielding School of Public Health, University of California, Los Angeles, California, United States of America, 4 Center for Public Health and Disasters, Fielding School of Public Health, University of California, Los Angeles, California, United States of America, 5 University of California Office of the President, Oakland, California, United States of America, 6 Department of Epidemiology and Biostatistics, University of California, San Francisco, California, United States of America, 7 Department of Health Society and Behavior, Program in Public Health, Department of Neurology, School of Medicine, Susan and Henry Samueli College of Health Sciences, University of California Irvine, Irvine, California, United States of America, 8 Student Health, University of California Santa Barbara, Santa Barbara, California, United States of America, 9 Division of Infectious Diseases and Global Public Health, Department of Medicine, University of California San Diego, San Diego, California, United States of America, 10 Division of Pediatric Infectious Diseases, Department of Pediatrics, University of California San Francisco, San Francisco, California, United States of America

¤ Current address: Sansum Clinic, Santa Barbara, California, United States of America
* bpollock@ucdavis.edu

## Abstract

### Background

Epidemics of COVID-19 in student populations at universities were a key concern for the 2020–2021 school year. The University of California (UC) System developed a set of recommendations to reduce campus infection rates. SARS-CoV-2 test results are summarized for the ten UC campuses during the Fall 2020 term.

### Methods

UC mitigation efforts included protocols for the arrival of students living on-campus students, non-pharmaceutical interventions, daily symptom monitoring, symptomatic testing, asymptomatic surveillance testing, isolation and quarantine protocols, student ambassador programs for health education, campus health and safety pledges, and lowered density of on-campus student housing. We used data from UC campuses, the UC Health–California Department of Public Health Data Modeling Consortium, and the U.S. Census to estimate the proportion of each campus' student populations that tested positive for SARS-CoV-2 and compared it to the fraction individuals aged 20–29 years who tested positive in their respective counties.

**Data Availability Statement:** All relevant data are within the manuscript.

**Funding:** Unfunded study. The authors received no specific funding for this work.

**Competing interests:** No authors have competing interests

## Results

SARS-CoV-2 cases in campus populations were generally low in September and October 2020, but increased in November and especially December, and were highest in early to mid-January 2021, mirroring case trajectories in their respective counties. Many students were infected during the Thanksgiving and winter holiday recesses and were detected as cases upon returning to campus. The proportion of students who tested positive for SARS-CoV-2 during Fall 2020 ranged from 1.2% to 5.2% for students living on campus and was similar to students living off campus. For most UC campuses the proportion of students testing positive was lower than that for the 20–29-year-old population in which campuses were located.

## Conclusions

The layered mitigation approach used on UC campuses, informed by public health science and augmented perhaps by a more compliant population, likely minimized campus transmission and outbreaks and limited transmission to surrounding communities. University policies that include these mitigation efforts in Fall 2020 along with SARS-CoV-2 vaccination, may alleviate some local concerns about college students returning to communities and facilitate resumption of normal campus operations and in-person instruction.

## Introduction

After rapidly moving classes online and sending students home at the onset of the COVID-19 pandemic, many colleges and universities developed plans to reduce transmission risk of severe acute respiratory syndrome coronavirus 2 (SARS-CoV-2) infection in preparation for students returning for the 2020–2021 academic term. As students resumed classes in Fall 2020, SARS-CoV-2 outbreaks were tracked and frequently reported in the media. For example, after identifying 177 students who tested positive during the first week of in-person instruction, the University of North Carolina Chapel Hill moved all undergraduate instruction to fully online [1] and when 16% of tests were positive in early August, the University of Notre Dame paused in-person instruction [2]. Even higher proportions of positive tests were observed at some institutions of higher education in 2021 than in 2020 [3].

For returning students living on campus, colleges and universities employed various testing, prevention, and other mitigation strategies. These commonly included pre-arrival or on-arrival screening of all students, daily symptom tracking, fixed serial asymptomatic testing, on-demand testing for symptomatic individuals, universal employee testing, isolation for those testing positive, and contact tracing and quarantine of high-risk contacts. Less common strategies included sentinel sampling of campus populations and wastewater surveillance of campus buildings. While attention initially focused on testing, more comprehensive approaches were required as testing alone is insufficient to prevent transmission without contact tracing, and timely isolation and quarantine. Other preventive measures include adequate provisioning and use of personal protective equipment, physical distancing, environmental management with attention to heating, optimization of air conditioning and ventilation systems, and a preference toward the use of outdoor rather than indoor spaces [4]. Model-based studies have found that implementing multiple approaches is most effective [5] and even cost-effective at controlling transmission among university students [6].

To facilitate safer reopening of the University of California (UC) System campuses, several workgroups were organized under the UC Office of the President (UCOP) and a coherent set of recommendations was developed aimed at reducing campus infection rates. The UC Health Coordinating Committee (HCC), chaired by the Executive Vice President of UC Health, convened several systemwide workgroups and task forces to lead the university response to the COVID-19 pandemic. These groups contributed subject matter expertise and prioritized response objectives such as conducting campus testing and contact tracing, ensuring sufficient campus testing capacity, and implementing other safety measures (UC System committee members are listed in S1 Appendix).

In May 2020, the UC Systemwide COVID-19 Public Health Workgroup developed guidance that included a range of recommendations in several broad areas including work environment distancing; use of classrooms and other instructional spaces; student housing; university common spaces such as libraries, dining facilities, and recreational spaces; student behavior and responsibility; and mental health and emotional support. Other UCOP HCC groups included the Campus Testing and Tracing Task Force, the Campus Testing Capacity Task Force, and the Symptom Screening Task Force. The product of these workgroups was distributed to leadership at each of the 10 UC campuses [7].

UC recommendations included lowering on-campus density and transitioning to predominantly remote instruction, streamlining symptomatic testing with 24 hour turn-around of results, requiring widespread asymptomatic testing, enhancing case finding and contact tracing, implementing exposure notification technology, providing isolation and quarantine facilities, developing student ambassador programs for campus health education, implementing a UC Systemwide influenza vaccination mandate, and building strong partnerships with local health authorities. While the independent evaluation of single recommendations is impractical, the aggregate impact of these layered preventive measures, measured by the proportion of SARS-CoV-2 tests that were positive, was assessed for the Fall 2020 quarter and semester.

This report summarizes the on-campus COVID-19 experience across a very large, geographically diverse university system and provides an indication of whether such measures can be useful for planning as the pandemic continues. We determined the proportion of students at UC campuses that tested positive for SARS-CoV-2 during the Fall 2020 term (quarter or semester) and compared this to the incidence in young adults 20–29 years in counties where campuses are located.

## Materials and methods

The University of California, among the largest U.S. public university systems, includes 10 campuses with 5 academic medical centers and 3 national laboratories. Fig 1 shows the geographic distribution of campuses; the national laboratories were not included in this analysis. In Fall 2020, total enrollment was 285,862 students, including 226,449 undergraduates and 59,267 graduate and professional students. Historically, this is the highest enrollment for the UC System ever compared with 285,216 students enrolled in Fall 2019 and 280,386 in Fall 2018. There were 216,200 UC System employees in October of 2020, lower than in previous years—236,052 in October 2019 and 229,214 in October 2018.

### UC systemwide campus guidance

A range of preventive interventions for the UC campuses were required or recommended and are summarized below [8].

**On-campus arrival protocols.** Asymptomatic SARS-CoV-2 testing was recommended for all students within 7 days prior to arrival on campus. If not available in the student's home

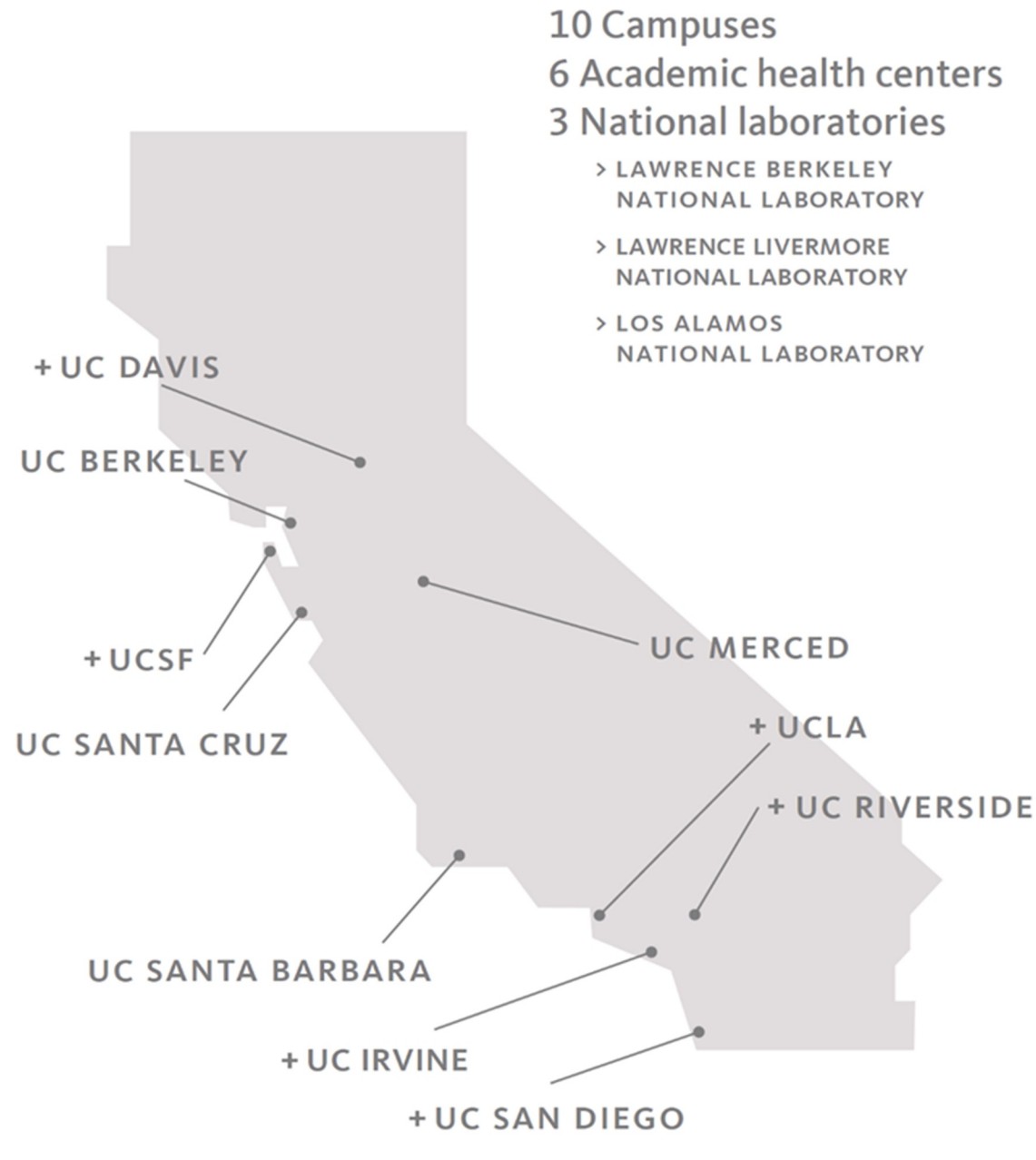

**Fig 1. Geographic location of University of California campuses.**

community, testing was performed upon arrival on campus. A second test was required within 7–14 days post-arrival (or more frequently based on an individual UC campus's asymptomatic testing requirements) for students residing on campus.

Campus residents were required to sequester for their first seven days. Sequestering was defined as minimizing in-person interactions among students, faculty, and staff whether in the dormitories, dining facilities, classrooms, or other locations where students congregate on or off campus. Students with COVID-19-related symptoms were asked not to travel to campus until they received a medical evaluation and care and were symptom-free for ≥72 hours.

**Non-pharmaceutical interventions.** Non-pharmaceutical interventions included but were not limited to wearing facial coverings while in public, practicing physical distancing (or the installation of easily cleaned physical barriers where physical distancing was not possible), performing frequent hand hygiene, avoiding crowds and poorly ventilated indoor spaces, and implementing appropriate cleaning and disinfection protocols. Food service was modified to take-out only rather than in-person dining.

**Daily symptom monitoring and symptomatic testing.** In the Fall 2020, all campuses implemented a process for screening university students, faculty, staff, and visitors (if allowed on a particular campus) for COVID-19 symptoms [9] prior to allowing access to any university facilities, including classroom and research buildings, dining halls, libraries, and congregate living facilities. Several campuses used symptom profiles to prioritize who received PCR tests and determine the frequency of testing.

**Periodic asymptomatic surveillance testing.** In addition to symptomatic testing for all employees and students, campuses adopted an on-campus SARS-CoV-2 screening program for periodic testing of asymptomatic individuals. Asymptomatic testing was made available for all onsite students. Staff and faculty had access to regular asymptomatic testing at some campuses. Some campuses also routinely tested students living off-campus as high density off-campus housing presents additional challenges for SARS-CoV-2 transmission containment. The frequency of asymptomatic testing required was based on local conditions, campus density, and testing capacity.

**Isolation and quarantine protocols.** Many of the UC campuses, in close collaboration with local health departments, stood up case identification/contact tracing teams that quickly identified potential contacts of PCR-positive cases and quarantined individuals according to CDC and state/local health policies. Designated dormitories served as temporary dwellings for students (and in some cases employees) who were found to be infected with SARS-CoV-2 or identified as potentially exposed. To avoid sending students to their home communities and possibly exposing others, they were provided lodging to safely isolate or quarantine on-campus in consultation with Student Health, Employee Health, and local health departments. Isolation rooms including private baths, limited kitchen facilities, and food service were provided to prevent spread to susceptible individuals. To assure continued academic productivity, computer equipment and Internet connections were made available, as well as instructional materials for students and tele-work capabilities for employees.

**Student ambassadors and campus health and safety pledge.** Campuses required residential students to sign a pledge, which was a commitment to help reduce the spread of COVID-19 on their campus. Some campuses developed student ambassador programs which used in-person encounters, social media, chatlines and other communication strategies to answer questions and provide updates to increase adherence to public health recommendations.

**Compliance with state and local health guidance and law.** As part of a larger effort to comply with Assembly Bill 685 and Cal/OSHA Emergency Temporary Standard Title 8 Section 3205, which require California employers to notify employees and their representatives of a potential exposure to SARS-CoV-2 [10], the UC academic campuses tracked COVID-19 testing and active case data. These data were compiled and displayed on publicly web-accessible campus COVID-19 dashboards providing a required "notice of potential exposure" to employees and their representatives (Table 1).

## Data analysis

Testing data were obtained from the UC campuses. U.S. Census as well as data from the University of California Health–California Department of Public Health data modeling

**Table 1. University of California campus COVID-19 dashboards.**

| Campus | | COVID-19 Dashboard URL |
|---|---|---|
| UCB | UC Berkeley | https://coronavirus.berkeley.edu/dashboard/ |
| UCD | UC Davis | https://campusready.ucdavis.edu/testing-response/dashboard |
| UCI | UC Irvine | https://uci.edu/coronavirus/dashboard/index.php |
| UCLA | UC Los Angeles | https://www.uclahealth.org/coronavirus |
| UCM | UC Merced | https://emergency.ucmerced.edu/covid19-dashboard |
| UCR | UC Riverside | https://ehs.ucr.edu/coronavirus/cases |
| UCSB | UC Santa Barbara | https://www.ucsb.edu/COVID-19-information/dashboard |
| UCSC | UC Santa Cruz | https://recovery.ucsc.edu/reporting-covid/covid-tracking/ |
| UCSD | UC San Diego | https://returntolearn.ucsd.edu/dashboard/index.html |
| UCSF | UC San Francisco | https://coronavirus.ucsf.edu/dashboard |

consortium [11] were used for analysis. We calculated the proportion of each campus' student populations that tested positive for SARS-CoV-2 during the fall term (quarter or semester) and compared this to the proportion of young adults (ages 20 to 29 years) testing positive in the county in which the campus was located. Age-specific test data were not available for Yolo County separately where the main UC Davis campus is located, so we used data for four grouped counties: Yolo, Yuba, Sutter, and El Dorado. Because no students or trainees are housed there, the UC Davis Medical Center campus in Sacramento County was not included in the analysis. Although the incidence for the county included the University cases, they were a small fraction of the cases among 20–29-year-olds (mean 6.3%, range 1–25%), and the University populations, including students living on-campus and off-campus, were an average of 6.7% of the county 20–29-year-old population (range 1–13%). For seven campuses (Berkeley, Davis, Irvine, Los Angeles, Santa Barbara, Santa Cruz, and San Diego) data were available to enable discrimination of COVID-19 cases for students living on- and off-campus. The number of students who returned to campus communities but lived off campus was not precisely known and was estimated including from pre-term student surveys.

SARS-CoV-2 testing information for the quarter or semester was gathered for each campus starting from the first day of instruction through the end of final examinations, ranging from August 19, 2021, to December 19, 2021.

We statistically compared the incidence rates for each campus population (on- and off-campus students) to the County 20–29-year-old incidence. A generalized linear model with a Poisson distribution was used with cases as the response variable, population as the fixed effect predictor, and the log of the person-days of exposure during the Fall term as an offset. Statistical significance was defined as $p < 0.05$ (2-sided).

## Results

The number and proportion of enrolled students who returned to live on campus in the Fall 2020 varied greatly across the UC System (Table 2). Of the universities with undergraduate students, UC San Diego, which implemented one of the first universal student testing programs in the U.S., had the highest proportion and largest number of students living on campus (23.1%, n = 9,129) followed by UC Irvine (19.8%, n = 7,182), and UC Davis (10.2%, n = 4,000). Estimates of the fraction of enrolled students living nearby but off campus ranged from 20% to 40% (except for UC San Francisco, with 82%). Collectively, the UC System's robust testing efforts resulted in 521,449 tests during the Fall term, including symptomatic testing with return of results within 24 hours.

**Table 2. Fall 2020 University of California campus and surrounding population sizes, SARS-CoV-2 testing volume, COVID-19 cases and fraction of populations testing positive during the fall term.**

| | UCB | UCD | UCI | UCLA | UCM | UCR | UCSB | UCSC | UCSD | UCSF |
|---|---|---|---|---|---|---|---|---|---|---|
| Fall 2020 Quarter/Semester enrollment[a] | 42,327 | 39,074 | 36,303 | 44,589 | 9,018 | 26,434 | 26,179 | 19,161 | 39,576 | 3,215 |
| Number of students living on-campus | 2,189 | 4,000 | 7,182 | 703 | 388 | 1,815 | 1,319 | 950 | 9,129 | 579 |
| Percent of students Living on campus | 5.2% | 10.2% | 19.8% | 1.6% | 4.3% | 6.9% | 5.0% | 5.0% | 23.1% | 18.1% |
| Number of Staff/Faculty on-campus during Fall[b] | 3,500 | 7,000 | 3,688 | 9,660 | 600 | 2,709 | 3,849 | 1,200[c] | 3,914 | 19,750 |
| Estimated number of students living locally off-campus | 8,000–10,000 | 16,000 | 10,704 | n.a. | 3,000 | 1,000 | 8,400 | 5,000 | 15,751 | 2,636 |
| Number of unique individuals tested | 17,111 | 24,290 | 8,478 | 22,357[d] | 2,026 | 2,153 | 3,139 | 5,637 | 25,391 | 11,766 |
| Number of tests collected | 96,284 | 86,648 | 54,070 | 103,077 | 5,804 | 15,532 | 16,268 | 26,015 | 94,811 | 22,672 |
| Number of positive tests collected | 264 | 202 | 403 | 694 | 161 | 63 | 68 | 97 | 424 | 413 |
| Number of positive cases in students | 235 | 145 | 313 | 227 | 109 | 59 | 53 | 79 | 307 | 22 |
| Percent of county 20–29-year-olds testing positive[e] | 2.4% | 3.1% | 3.7% | 5.0% | 5.1% | 5.6% | 1.9% | 2.4% | 3.5% | 1.6% |
| Percent of on-campus students testing positive | 2.1% | 4.7% | *2.0%* | 4.8% | 5.2% | *3.3%* | 4.6% | *1.4%* | *1.2%* | 1.7% |
| Percent of off-campus students testing positive | *2.1%* | n.a. | *1.4%* | 4.1% | 2.0% | n.a. | 4.0% | *1.2%* | *1.2%* | *0.4%* |
| Academic term | Semester | Quarter | Quarter | Quarter | Semester | Quarter | Quarter | Quarter | Quarter | Quarter |
| Start date[f] | 8/19/20 | 9/28/20 | 9/28/20 | 9/28/20 | 8/19/20 | 9/27/20 | 9/26/20 | 9/28/20 | 9/28/20 | 9/28/20 |
| End date[f] | 12/18/20 | 12/18/20 | 12/18/20 | 12/18/20 | 12/18/20 | 12/18/20 | 12/18/20 | 12/18/20 | 12/19/20 | 12/18/20[g] |

[a]Enrollment numbers from https://www.universityofcalifornia.edu/infocenter/fall-enrollment-glance

[b]Number of faculty/staff on campus is estimated and excludes those on separate medical campuses except UC San Francisco

[c]Derived from the daily symptom screening tool

[d]Does not include unique individuals tested from commercial lab

[e]Italicized values in the next two rows show significantly higher and underlined values show significantly lower proportion testing positive from the Poisson regression (p<0.05, 2-sided)

[f]Academic calendar dates from published university registrar calendars

[g]End date is last day of final exams to maintain consistency with other campuses. n.a. = not available.

We examined the temporal patterns of the fraction of campus populations testing positive for seven UC campuses with available daily case data, and we extended the date range to examine the impact of arrival periods in August and September 2020 as well as post-winter holidays return to campuses. Cases were low in September and October 2020 usually, but increased in November and especially December. Test positivity proportions peaked in early to mid-January (Fig 2) upon return from the 2020 winter holidays. This mirrored case trajectories in the counties, which also showed a large increase from low numbers in early fall to much higher numbers from November to January (Fig 2). A sizeable number of students were infected during the winter holiday recess and were detected as cases in the first week or two in January (Fig 2). There was also a smaller, but observable, rise in cases detected following the Thanksgiving break at six of seven campuses (Fig 2; all but UC Santa Barbara).

The proportion of people 20 to 29 years old in counties with UC campuses that tested positive for SARS-CoV-2 during the fall term ranged from 1.6% (UC San Francisco) to 5.6% (UC Riverside) (Table 2). The proportion of students living on campus that tested positive for SARS-CoV-2 during this period ranged from 1.2% to 5.2% and was similar for students living off campus (Table 2). The proportion of on-campus students testing positive for SARS-CoV-2 was lower than or equal to that for the 20 to 29-year-old population in all counties where campuses were located except UC Santa Barbara and UC Davis (Table 2).

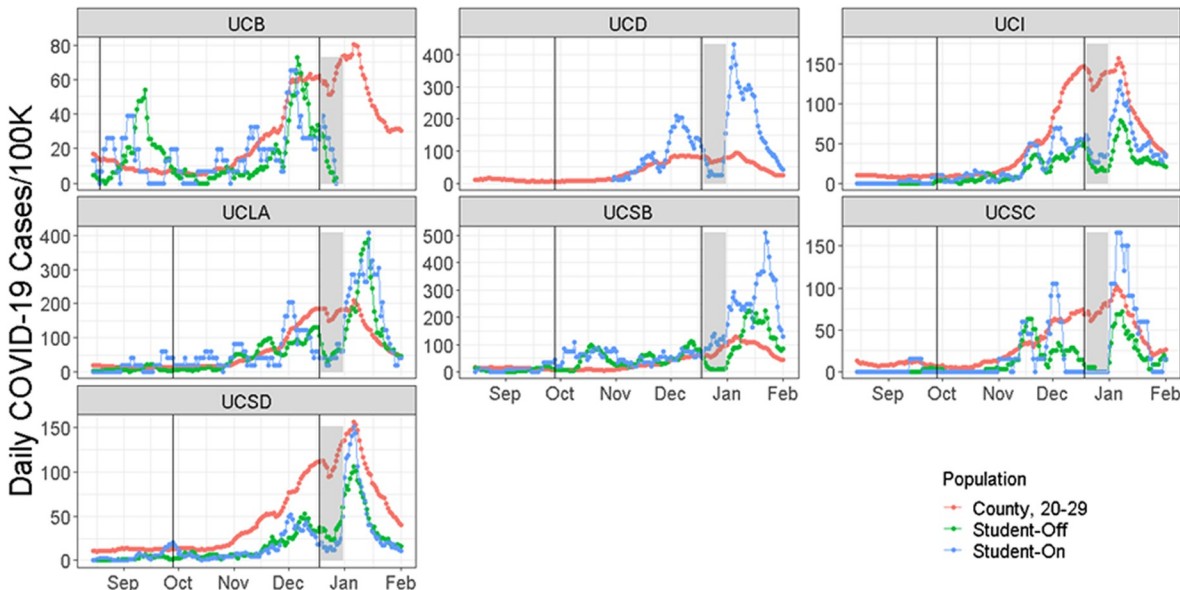

**Fig 2. The 7-day running average of the daily number of people testing positive for SARS-CoV-2 per 100,000 individuals over time between August 15, 2020, and February 1, 2021.** Vertical lines show the start and end of the fall quarter or semester at these seven UC campuses, which is the period used for calculating the fraction of each population testing positive in Table 2. The grey bars show the winter break period when fewer students were on campus and being tested. For UCD, daily data were only available for asymptomatic testing of on-campus students and only after 10/27/20. For UCB data were available through 12/30/2020.

## Discussion

Institutions of higher education have deployed several strategies to mitigate on-campus COVID-19 outbreaks. In an ecologic study, strong evidence of reduced community transmission emerged for counties with large universities (≥20,000) that transitioned to remote instruction compared with universities that did not [12] Lowered housing density, mandated use of non-pharmaceutical interventions, and traditional epidemiologic infectious disease control measures (testing, contact tracing, and quarantine/isolation) all contribute to safer campus environments. The collective impact of deploying these measures has not been well delineated at other universities.

There are several important limitations for this analysis. Comparisons were ecologic and relied on numbers based on geographic aggregation across entire campuses and even greater aggregation across whole counties, thus results are subject to the ecological inference fallacy. Cumulative positivity proportions over the entire quarter or semester for some campuses were used, as daily values were not available for every campus. Variability was likely due to each campus deploying their own testing procedures and platforms which ranged from rapid antigen to viral PCR tests with nasopharyngeal swabs, nasal swabs, or saliva-based specimens. Results from both symptomatic and asymptomatic testing were included from campuses and could not be disaggregated. There were differences across campuses in testing schedules (including more frequent testing for certain groups such as student athletes), as well variability related to adherence to required testing. Misclassification of time-at-risk may have occurred as students and employees were not tracked for actual time spent on campus. For campus comparisons to counties, age-specific test data were only available in 10-year increments; county age ranges of 20–29 years only partly overlap the age range of university students (18–23, with younger students being more likely to be on-campus students). Finally, aggregated campus test results precluded adjustment for potential confounding or effect modification.

Strengths of this analysis include assessment of one of the largest university systems in the U.S. Campuses, which vary in size, are dispersed geographically, and located in counties with different outbreak dynamics. Also, UC campuses had to comply with mandated state reporting regulations on potential exposure to SARS-CoV-2 along with required notification to local county health authorities and the State of California's reporting database, the California Reportable Disease Information Exchange (CalREDIE). Finally, because of mandated asymptomatic testing programs on the UC campuses, it would be expected that campus cases were more likely to be identified than cases from the rest of the county, providing a higher level of certainty that case rates were actually lower at the majority of UC campuses.

In this analysis of SARS-CoV-2 attack rates, we found that the proportions of students that tested positive for the 10 UC campuses were mostly lower than those in the counties in which they are located although the magnitude varied across campuses except for UC Santa Barbara and UC Davis. However, UC Davis implemented a comprehensive community-wide COVID-19 control program that generally lowered test positivity in the county possibly accounting for less of a difference with on-campus positivity rates [13]. Given our results, it appears unlikely that UC campuses served as a major drivers of community transmission, as may have occurred in other states [14].

The layered mitigation approach, informed by public health science, likely minimized campus transmission and outbreaks. These strategies, if widely deployed, may alleviate local concerns about unvaccinated college students returning to communities and spreading contagion. Screening students returning to campus at the start of the term and after the Thanksgiving and winter holidays (when incidence in California was elevated) was a key strategy to detect infections in students and likely reduced subsequent transmission.

This analysis suggests that students living on or near UC campuses were usually at lower risk than young adults in the surrounding population. Our results also accord with on-campus modeling work at other universities [14–16] and provide support for integrated public health efforts targeted at an entire university system to prevent the spread of SARS-CoV-2.

For institutions of higher education making plans for Fall 2021, the policy implications of our findings strongly suggest that repeated SARS-CoV-2 testing be mandated for all students and employees physically on campus, that non-pharmaceutical interventions be required and monitored, that sufficient case identification/contact tracing workforce be available, and that adequate space for isolation and quarantine be available. The addition of SARS-CoV-2 vaccination is expected to make Fall 2021 much safer than Fall 2020 and allow for increased density. As such, the University of California System has implemented a vaccine mandate. There will also likely be emerging evidence that wastewater monitoring and other environmental monitoring will augment campus infectious disease control measures. Combined, these efforts may alleviate some local concerns about college students returning to communities and spreading contagion, as well as facilitate resumption of normal campus operations and in-person instruction.

## Supporting information

**S1 Appendix. Groups that have contributed to the University of California system-wide COVID-19 efforts.**
(DOCX)

## Acknowledgments

This report represents contributions of many individuals in the University of California System, listed in S1 Appendix.

## Author Contributions

**Conceptualization:** Brad H. Pollock, A. Marm Kilpatrick, David P. Eisenman, Kristie L. Elton, George W. Rutherford, Bernadette M. Boden-Albala, David M. Souleles, Laura E. Polito, Natasha K. Martin, Carrie L. Byington.

**Data curation:** Brad H. Pollock, A. Marm Kilpatrick, Kristie L. Elton.

**Formal analysis:** Brad H. Pollock, A. Marm Kilpatrick.

**Investigation:** Brad H. Pollock, David P. Eisenman, Kristie L. Elton, George W. Rutherford, Bernadette M. Boden-Albala, David M. Souleles, Laura E. Polito, Natasha K. Martin, Carrie L. Byington.

**Methodology:** Brad H. Pollock, A. Marm Kilpatrick.

**Project administration:** Brad H. Pollock, Kristie L. Elton.

**Supervision:** Brad H. Pollock.

**Visualization:** A. Marm Kilpatrick.

**Writing – original draft:** Brad H. Pollock, A. Marm Kilpatrick, David P. Eisenman, Kristie L. Elton, George W. Rutherford, Bernadette M. Boden-Albala, David M. Souleles, Laura E. Polito, Natasha K. Martin, Carrie L. Byington.

**Writing – review & editing:** Brad H. Pollock, A. Marm Kilpatrick, David P. Eisenman, Kristie L. Elton, George W. Rutherford, Bernadette M. Boden-Albala, David M. Souleles, Laura E. Polito, Natasha K. Martin, Carrie L. Byington.

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
