## [Decision Letter · Decision Letter 0]

18 Aug 2021

PONE-D-21-19364

Safe reopening of college campuses in Fall 2020: The University of California experience

PLOS ONE

Dear Dr. Pollock,

Thank you for submitting your manuscript to PLOS ONE. After careful consideration, we feel that it has merit but does not fully meet PLOS ONE’s publication criteria as it currently stands. Therefore, we invite you to submit a revised version of the manuscript that addresses the points raised during the review process.

We look forward to receiving your revised manuscript.

Kind regards,

Dong Keon Yon, MD, FACAAI

Academic Editor

PLOS ONE

Journal Requirements:

 "Unfunded study"

3. We note that Figure (1)  in your submission contain map images which may be copyrighted. All PLOS content is published under the Creative Commons Attribution License (CC BY 4.0), which means that the manuscript, images, and Supporting Information files will be freely available online, and any third party is permitted to access, download, copy, distribute, and use these materials in any way, even commercially, with proper attribution. For these reasons, we cannot publish previously copyrighted maps or satellite images created using proprietary data, such as Google software (Google Maps, Street View, and Earth). For more information, see our copyright guidelines: http://journals.plos.org/plosone/s/licenses-and-copyright.

1. You may seek permission from the original copyright holder of Figure (1) to publish the content specifically under the CC BY 4.0 license.  

Additional Editor Comments (if provided):

Thank you for considering for publication in the Plos One.

Reviewers' comments:

Reviewer's Responses to Questions

**Comments to the Author**

1. Is the manuscript technically sound, and do the data support the conclusions?

Reviewer #1: Yes

Reviewer #2: Yes

2. Has the statistical analysis been performed appropriately and rigorously? 

Reviewer #1: N/A

Reviewer #2: Yes

3. Have the authors made all data underlying the findings in their manuscript fully available?

Reviewer #1: Yes

Reviewer #2: Yes

4. Is the manuscript presented in an intelligible fashion and written in standard English?

Reviewer #1: Yes

Reviewer #2: Yes

5. Review Comments to the Author

Reviewer #1: Safe reopening of college campuses in Fall 2020: The University of California experience

Original Submission

Doaa M. Abdel-Salam, Associate Professor

The manuscript is technically sound. I recommend acceptance.

Reviewer #2: # 1. Please revise the title of your manuscript in more attractive.

i.e. Safe reopening of college campuses during COVID-19 pandemic: The University of California experience in Fall 2020

#2. Please suggest a paragraph of policy implications including vaccine effects, and a professional biological mechanism.

6. PLOS authors have the option to publish the peer review history of their article (what does this mean?). If published, this will include your full peer review and any attached files.

Reviewer #1: No

Reviewer #2: No

---

## [Author Response · Author response to Decision Letter 0]

28 Sep 2021

September 28

Dong Keon Yon, MD, FACAAI,

RE: Response PLOS ONE Review of PONE-D-21-19364

Dear Dr. Yon:

Thank you so much for your review of our manuscript. We are submitting a revised manuscript that addresses the reviewers’ comments. 

Responses to the Journal Requirement Comments:

1. We reviewed the journal requirements and believe that the current submission meets all of those requirements.

2. We have provided additional text for the Financial Disclosure that states, “Unfunded study. The authors received no specific funding for this work.” 

3. The map that is included as Figure 1 was not produced using copyrighted material of any kind. It was hand drawn by the University of California Office of the President and is not copyrighted. 

Responses to the Reviewers’ Comments: 

5. #2 Please revise the title of your manuscript in more (sic) attractive. 

We have revised the title as suggested by Reviewer #2. 

 #2 Please suggest a paragraph of policy implications including vaccine effects, and a professional biological mechanism. 

We agree with Reviewer #2 and feel that a more explicit policy implication statement is an appropriate addition. Therefore, we have modified the last paragraph of the Discussion section (and the abstract) and added this statement. We do not understand the “professional biological mechanism” part of this comment as this is a population-based analysis. 

Sincerely,

Brad H. Pollock, M.P.H., Ph.D., F.A.C.E.

Distinguished Professor and Chairman,

Associate Dean for Public Health Sciences, 

School of Medicine

---

## [Decision Letter · Decision Letter 1]

5 Oct 2021

Safe reopening of college campuses during COVID-19: The University of California experience in Fall 2020

PONE-D-21-19364R1

Dear Dr. Pollock,

We’re pleased to inform you that your manuscript has been judged scientifically suitable for publication and will be formally accepted for publication once it meets all outstanding technical requirements.

Kind regards,

Dong Keon Yon, MD, FACAAI

Academic Editor

PLOS ONE

Additional Editor Comments (optional):

i congratulate you on this mermerzing work.

Reviewers' comments:

Reviewer's Responses to Questions

**Comments to the Author**

1. If the authors have adequately addressed your comments raised in a previous round of review and you feel that this manuscript is now acceptable for publication, you may indicate that here to bypass the “Comments to the Author” section, enter your conflict of interest statement in the “Confidential to Editor” section, and submit your "Accept" recommendation.

Reviewer #2: All comments have been addressed

2. Is the manuscript technically sound, and do the data support the conclusions?

Reviewer #2: Yes

3. Has the statistical analysis been performed appropriately and rigorously? 

Reviewer #2: Yes

4. Have the authors made all data underlying the findings in their manuscript fully available?

Reviewer #2: Yes

5. Is the manuscript presented in an intelligible fashion and written in standard English?

Reviewer #2: Yes

6. Review Comments to the Author

Reviewer #2: #2 Please revise the title of your manuscript in more (sic) attractive.

#2 Please suggest a paragraph of policy implications including vaccine effects, and a professional biological mechanism.

I have no further edits.

7. PLOS authors have the option to publish the peer review history of their article (what does this mean?). If published, this will include your full peer review and any attached files.

Reviewer #2: No

---

## [Editor Report · Acceptance letter]

26 Oct 2021

PONE-D-21-19364R1 

Safe reopening of college campuses during COVID-19: The University of California experience in Fall 2020 

Dear Dr. Pollock:

I'm pleased to inform you that your manuscript has been deemed suitable for publication in PLOS ONE. Congratulations! Your manuscript is now with our production department. 

Kind regards, 

on behalf of

Dr. Dong Keon Yon 

Academic Editor

PLOS ONE